# Properties of multivesicular release from mouse rod photoreceptors support transmission of single-photon responses

Cassandra L Hays[1,2], Asia L Sladek[3], Greg D Field[4]*, Wallace B Thoreson[1,3]*

[1]Truhlsen Eye Institute and Department of Ophthalmology and Visual Sciences, College of Medicine, University of Nebraska Medical Center, Omaha, United States; [2]Cellular and Integrative Physiology, Omaha, United States; [3]Pharmacology and Experimental Neuroscience, Omaha, United States; [4]Department of Neurobiology, Duke University School of Medicine, Durham, United States

**Abstract** Vision under starlight requires rod photoreceptors to transduce and transmit single-photon responses to the visual system. Small single-photon voltage changes must therefore cause detectable reductions in glutamate release. We found that rods achieve this by employing mechanisms that enhance release regularity and its sensitivity to small voltage changes. At the resting membrane potential in darkness, mouse rods exhibit coordinated and regularly timed multivesicular release events, each consisting of ~17 vesicles and occurring two to three times more regularly than predicted by Poisson statistics. Hyperpolarizing rods to mimic the voltage change produced by a single photon abruptly reduced the probability of multivesicular release nearly to zero with a rebound increase at stimulus offset. Simulations of these release dynamics indicate that this regularly timed, multivesicular release promotes transmission of single-photon responses to post-synaptic rod-bipolar cells. Furthermore, the mechanism is efficient, requiring lower overall release rates than uniquantal release governed by Poisson statistics.

**\*For correspondence:**
field@neuro.duke.edu (GDF);
wbthores@unmc.edu (WBT)

**Competing interests:** The authors declare that no competing interests exist.

## Introduction

One of the most impressive features of the visual system is the ability to detect single photons. Pioneering psychophysical studies showed that humans can detect flashes consisting of a few photons hitting the retina, indicating that individual rods can respond to the absorption of single photons and then reliably signal these events to post-synaptic neurons (*Barlow, 1956*; *Hecht et al., 1942*; *Sakitt, 1972*; *Tinsley et al., 2016*). The ability of rods to respond to single photons was subsequently confirmed by electrophysiological recordings (*Baylor et al., 1984*; *Gross et al., 2015*; *Reingruber et al., 2015*; *Rieke and Baylor, 1998*). In darkness, rods maintain a relatively depolarized membrane potential of around −40 mV, and the absorption of a photon produces a small hyperpolarization, between 1 and 3.5 mV (*Cangiano et al., 2012*; *Hornstein et al., 2005*; *Schneeweis and Schnapf, 1995*). Rods signal the absorption of a photon to post-synaptic rod-bipolar cells by decreasing the rate of glutamate release. However, synaptic vesicle release is an intrinsically noisy process that is typically described by Poisson statistics (*Malagon et al., 2016*; *Miki, 2019*; *Zhang and Peskin, 2015*), posing a problem for post-synaptic bipolar cells to distinguish a genuine decrease in vesicle release caused by the absorption of a photon from a stochastic pause in vesicle release. How does the rod transform a small voltage change into a sufficiently large and reliable change in vesicle release?

One proposed solution is for a rod to maintain a high basal rate of vesicle release. Under the assumption of Poisson release, a rod would need to maintain a basal release rate of ~100 vesicles/s to generate a sufficiently reliable signal to distinguish single-photon responses from stochastic

variability in vesicle release (*Rao-Mirotznik et al., 1998*; *Rao et al., 1994*; *Schein and Ahmad, 2005*; *van Rossum and Smith, 1998*). This solution requires an extremely high turnover of vesicles that is energetically and physically demanding. Furthermore, measurements from salamander, gecko, and mouse rods show basal release rates of only 10–20 vesicles/s (*Hays et al., 2020b*; *Sheng et al., 2007*).

A second proposed solution is to make release more regular than a Poisson process. For example, implementing an Erlang process that releases vesicles at regular intervals would reduce the likelihood of mistaking a stochastic decrease in vesicle release for the absorption of a photon (*Schein and Ahmad, 2005*). Erlang events are generated by accumulating a certain number of Poisson events, which reduces interevent variability to regularize release. While Poisson statistics accurately describe vesicle release at many central synapses, this may not necessarily be true at the ribbon-style synapses in rods. Ribbons are plate-like, pre-synaptic structures that tether multiple vesicles along the surface and it has been suggested that delivery of vesicles down a ribbon may help to make release more regular (*Schein and Ahmad, 2005*).

To identify how single-photon responses can be signaled by changes in vesicle release from rods, we recorded pre-synaptic anion currents associated with glutamate transporter activity in mouse rod terminals (*Hays et al., 2020b*). The glutamate transporters in rods (largely EAAT5) are linked to an uncoupled anion conductance, so that an anion channel opens as glutamate is retrieved (*Arriza et al., 1997*; *Schneider et al., 2014*). Glutamate reuptake into rods continues even after post-synaptic mGluR6 receptors in rod-bipolar cells become saturated, suggesting that transporters are not saturated during rod release (*Hasegawa et al., 2006*). In that range, glutamate anion currents ($I_{A(glu)}$) vary linearly with glutamate levels in the synaptic cleft and can thus provide a pre-synaptic measure of glutamate release (*Hasegawa et al., 2006*; *Otis and Jahr, 1998*). Using these techniques, we observed stochastic release of individual vesicles that obeyed Poisson statistics when rods were voltage-clamped at −70 mV. However, at the typical resting membrane potential in darkness of −40 mV, release transitioned to regularly spaced multivesicular release events of ~17 vesicles apiece (*Hays et al., 2020b*). We found that the probability of initiating multivesicular release events was nearly abolished by a small membrane hyperpolarization similar to single-photon responses. A model of the empirically determined, voltage-dependent vesicle release dynamics combined with an ideal observer analysis reproduced the detection sensitivity of mouse rod-bipolar cells (*Field and Rieke, 2002a*). These results show that regularly timed multivesicular release events in rods can enhance post-synaptic detection of small voltage changes evoked by the absorption of individual photons. Furthermore, this strategy is efficient, achieving greater sensitivity than Poisson release while requiring fewer vesicles per second.

## Results

To study voltage-dependent vesicular glutamate release from rods, we recorded $I_{A(glu)}$ from rods using flatmount mouse retina preparations (*Figure 1*). When rods were held for many seconds at −60 or −70 mV, similar to the membrane potential achieved in bright light, we observed occasional inward currents arising from activation of glutamate transporters on the rod terminal (*Figure 1*; *Grassmeyer et al., 2019*). As described earlier (*Hays et al., 2020b*), these inward currents had waveforms typical of quantal post-synaptic currents but with a slow time course (10–90% rise time of 9–10 ms with decay time constants of ~40 ms). Spontaneous inward currents in rods voltage-clamped at −70 mV showed a unimodal amplitude distribution suggesting they consisted entirely of uniquantal events (*Figure 1C*). The overall rate of release increased at depolarized membrane potentials but achieved a rate at room temperature of only 4–5 vesicles/s at −40 mV, similar to the membrane potential in darkness. As shown previously, release rates and $I_{Ca}$ were doubled by increasing the bath temperature to 35°C, attaining an average rate of ~11 v/s/ribbon at −40 mV (*Hays et al., 2020b*). Accompanying this increased rate of vesicle release was a switch to more coordinated forms of release with semi-regular multivesicular release events dominating release in rods held at −40 mV (*Figure 1A,B*). Occasional unitary events were interspersed among these multivesicular events and so the overall distribution of event amplitudes measured at −40 mV was bimodal (*Figure 1C*). As described previously (*Hays et al., 2020b*), multivesicular release events involved the rapid sequential release of 10–20 vesicles triggered by $Ca^{2+}$ influx and derived from a pool of ribbon-associated vesicles shared with evoked release.

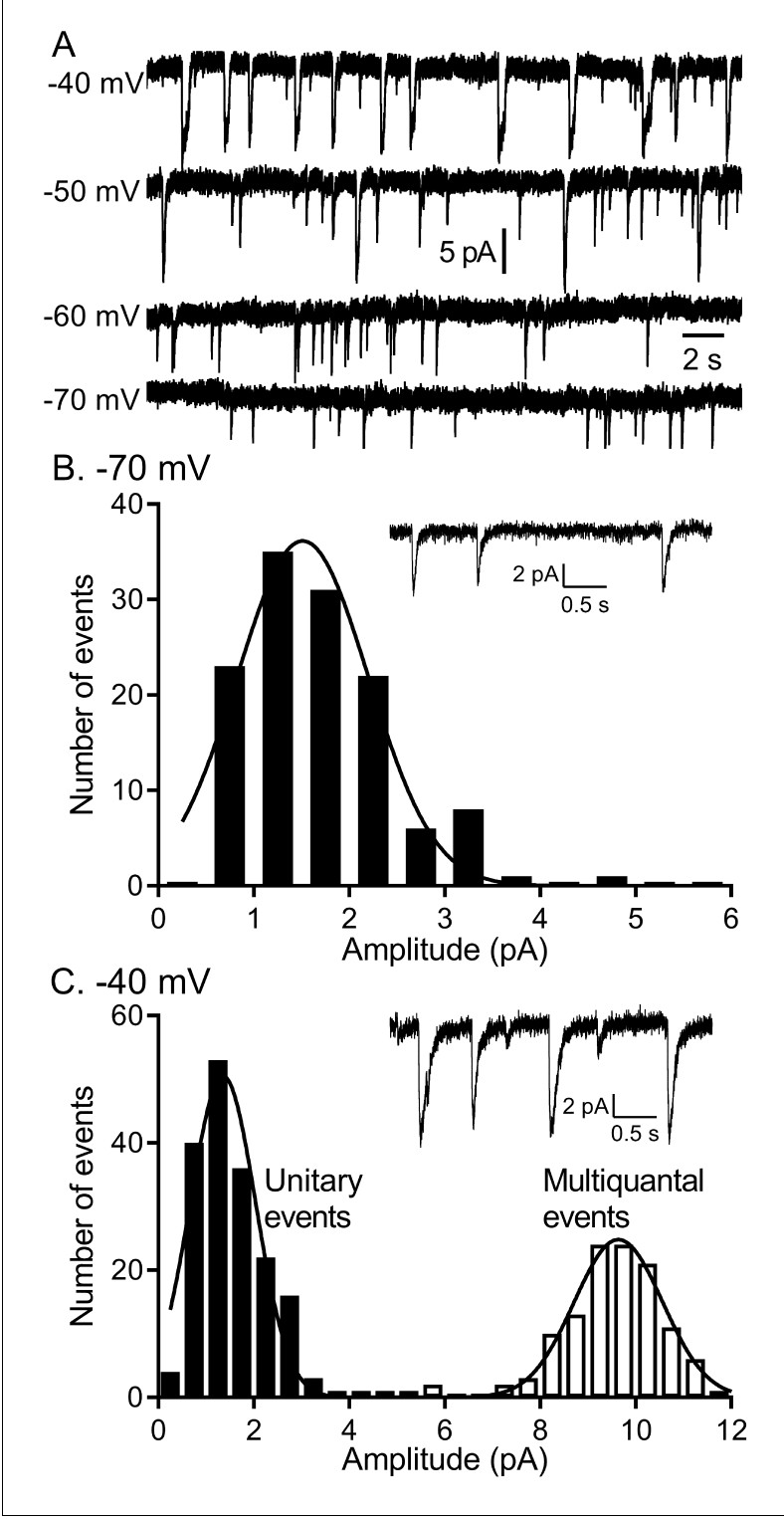

**Figure 1.** The quantity of glutamate released measured by $I_{A(glu)}$ increased with depolarization but near −40 mv, release shifted from single vesicle fusion events to multivesicular release events consisting of 10–20 vesicles apiece. (**A**) Representative traces from a single rod clamped at various voltages. (**B**) Example recording and amplitude histogram of unitary $I_{A(glu)}$ events from a rod voltage-clamped at −70 mV. Amplitude histogram was fit with a Gaussian function (mean ± SD = 1.51 ± 0.688 pA; n = 127 events). (**C**) Example recording and amplitude histogram from the same rod held at −40 mV. Amplitude histograms for unitary (black bars) and multivesicular

*Figure 1 continued on next page*

*Figure 1 continued*

(open bars) events were each fit with Gaussian functions (mean unitary event amplitude = 1.35 ± 0.685 pA; n = 178 events; multiquantal event amplitude = 9.64 ± 0.924 pA, n = 118 events).

## Multivesicular release was not modeled by Poisson statistics

It is generally accepted that under conditions of low release probability (e.g., when a neuron is hyperpolarized), vesicle exocytosis occurs stochastically, obeying Poisson statistics (*Malagon et al., 2016*; *Miki, 2019*; *Zhang and Peskin, 2015*). Spontaneous release can arise from $Ca^{2+}$-independent fusion events or from chance openings of voltage-gated $Ca^{2+}$ channels (*Cork et al., 2016*; *Kavalali, 2015*; *Kavalali, 2020*). The persistence of spontaneous events in rods held at −70 mV, even after blocking $Ca^{2+}$ channels with extracellular $Cd^{2+}$ or strongly buffering intracellular $Ca^{2+}$ with 10 mM BAPTA, showed evidence for $Ca^{2+}$-independent release (*Hays et al., 2020b*). To analyze frequency distributions, only rods in which we observed >50 events were included in the data set. The frequency distributions of interevent intervals for release events detected in rods voltage-clamped at −70 mV (e.g., *Figure 2A*) consistently exhibited a single exponential decay as predicted for a Poisson release process ($R^2$ = 0.97 ± 0.04, mean ± SD, n = 9 rods; *Figure 2C*). The probability of the number of release events that occur in a 1 s interval was also well fit using a Poisson probability distribution (*Equation 6*; *Figure 2E*). The Poisson fit for spontaneous events was robust ($R^2$ = 0.94 ± 0.07, n = 9 rods), and the mean number of events per second predicted from the Poisson fit, λ = 1.06 ± 0.55 (n = 9), was very close to the actual mean number of events observed in the same sample (λ = 1.0 ± 0.6, p=0.83, n = 9).

These results indicate that the rate of $Ca^{2+}$-independent spontaneous release from rods at hyperpolarized membrane potentials is a Poisson process. Does Poisson release persist when $Ca^{2+}$-dependent release is engaged at more depolarized potentials? At the typical resting potential in darkness (−40 mV), we observed periodic multivesicular events along with occasional unitary events (*Figure 2B*). Like those observed at −70 mV, the intervals between unitary events at −40 mV were well described by a single exponential (*Figure 2D*, filled squares, $R^2$ = 0.90 ± 0.07, n = 9 rods), consistent with an underlying Poisson release process. However, if we treated each multivesicular event as an individual release event, histograms of interevent intervals could not be fit by single exponentials, suggesting that multivesicular events are not released by a Poisson process (*Figure 2D*, bars). It therefore appears that single and multivesicular release events emerge independently.

We estimated the number of vesicles released in each event by dividing the total charge transfer during each multivesicular event by the average charge transfer for unitary events measured at the same potential in the same cell. Glutamate transporter currents continue to increase even after saturation of post-synaptic mGluR6 receptors, indicating that transporters are not normally saturated during release from rods (*Hasegawa et al., 2006*). This implies that the charge transfer for each $I_A$ (glu) vesicle release event should sum linearly. A Poisson distribution could not describe the frequency distribution of individual vesicle release events counted in this way. During each 1 s interval, there was either a multivesicular release event or no release event and so interevent intervals were distributed bimodally (*Figure 2F*). The frequency distribution could not be fit with a Poisson model (*Equation 6*), yielding negative $R^2$ values (−0.7 ± 0.6, n = 9) both when λ was unconstrained (λ = 2.5 in *Figure 2F*) and when λ was constrained to the actual mean number of events/s (λ = 11 in *Figure 2F*). Thus, unlike unitary event statistics, the statistics of multivesicular release were not well described by a Poisson distribution.

Fitting the interevent intervals for multivesicular events with a Gaussian function yielded better fits (mean $R^2$ = 0.84 ± 0.12) than fitting with an exponential function. Interevent intervals averaged 2.34 ± 0.64 s (N = 16) with a coefficient of variation (CV) among these cells of 0.43 ± 0.09. For a Poisson distribution where variance = mean, the average mean intervals between multivesicular events in this sample of cells predicted a CV of 0.67. Narrowing the interval distribution is a characteristic of an Erlang distribution that waits a certain number of Poisson intervals before each Erlang event. The number of Poisson intervals, k, that occur before each Erlang event is related to the CV by the formula:

$$CV = 1/(k)^{1/2}$$

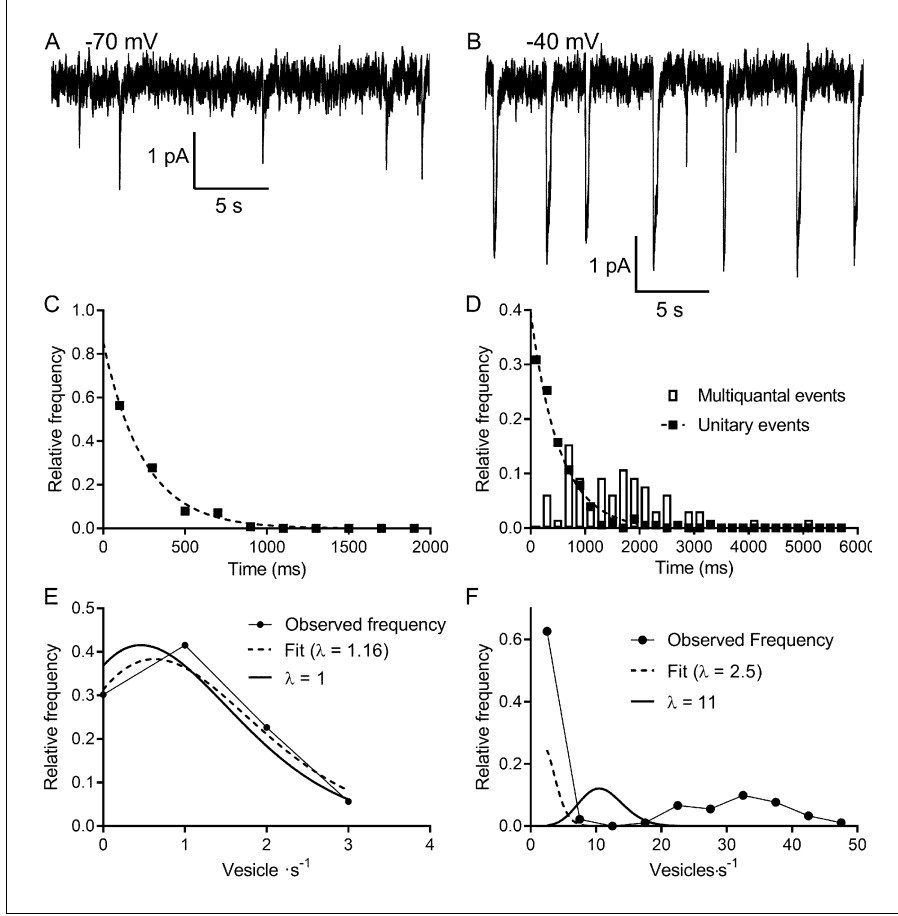

**Figure 2.** Multivesicular release events are not Poisson distributed in time. (**A, B**) Lengthy recording segments illustrating the stochastic release of individual vesicles at −70 mV (**A**) with more regular multivesicular release events at −40 mV (**B**). (**C**) Plot of the interevent interval frequency distribution for unitary events measured at −70 mV. Consistent with a Poisson distribution, these data were well fit with a one-phase exponential decay ($\tau$ = 248 ms; $R^2$ = 0.993). (**D**) Frequency histogram of intervals between both unitary (filled squares) and multivesicular (open bars) events. The distribution of intervals between unitary events was well described by a single exponential function ($\tau$ = 532 ms; $R^2$ = 0.985), whereas the intervals between multivesicular events were poorly fit with a single exponential. (**E**) Graph of the relative frequency of individual vesicle release events measured at −70 mV occurring in 1 s bins measured over 90–180 s. Data were fit with the Poisson equation (***Equation 6***) constrained to the observed mean of $\lambda$ = 1 event per s (dashed line, calculated, $R^2$ = 0.88). We obtained an even better fit but with a similar value for $\lambda$ when it was unconstrained (dashed line, fit $\lambda$ = 1.16, $R^2$ = 0.95). Data from the same cell shown in (**C** and **D**). (**F**) Graph of the relative frequency of quanta released in each 1 s bin. If we consider each of the quanta released in a multivesicular event as separate release events, then there were either 0 or ~35 quanta released per second. This distribution was not well described by Poisson statistics, whether $\lambda$ was constrained to match the actual mean quantal rate of 10.98 (solid line, $R^2$ = −0.39) or when $\lambda$ was unconstrained (dashed line, best fit $\lambda$ = 2.5, $R^2$ = 0.46).

The intervals between multivesicular events were consistent with Erlang factors ranging from 3 to 12 and averaged k = 6.2 ± 3.12 (n = 16).

## Impact of experimental conditions on statistics of multivesicular release

The conditions of our experiments differ from dark-adapted conditions in vivo. We therefore investigated how different aspects of these experiments might have impacted the observed release dynamics. We began by examining the impact of dark adaptation. During dark adaptation, ribbons can increase in length in mouse rods (***Dembla et al., 2020***) and bursts involve release from the pool of vesicles at the base of the ribbon (***Hays et al., 2020b***). Thus, dark adaptation might influence the

statistics of vesicle release. However, when holding dark-adapted rods at −40 mV, we found that the number of vesicles in each multivesicular release event (16.7 ± 5.8, n = 7) and the intervals between multivesicular event start times (2.0 ± 0.54 s, n = 6, room temperature) did not differ significantly from those measured in retinas prepared in the light (17 ± 7 v/burst, n = 22, p=0.74, unpaired t-test; 2.3 ± 0.64 s intervals, n = 16, p=0.34, unpaired t-test) (*Hays et al., 2020b*). Thus, recording from rods in light-adapted versus dark-adapted states did not strongly alter release statistics. One caveat is that in both light- and dark-adapted retinas, we mechanically removed outer segments to expose the inner segments and cell bodies prior to recording.

Next, we examined the potential impact of physiological rundown following membrane rupture by the patch pipette for voltage-clamping the rods. As illustrated in *Figure 3*, multivesicular events increased in amplitude during the first few minutes of recording as SCN⁻ diffused into the rod from the patch pipette. There was also a decrease in multivesicular event frequency after patch rupture, presumably due to rundown of the release mechanism. When the intervals between multivesicular events were plotted against time after patch rupture, the slopes of the linear regressions were all significantly non-zero (n = 5 rods, room temperature). The y-intercept values averaged 1.23 ± 0.45 s, whereas the overall mean interval between multivesicular events in this sample averaged 2.26 s ± 0.60 s (n = 5, room temperature). This suggests that the interevent intervals present at patch rupture were roughly half the overall average interval between multivesicular release events yielding approximately twice the release rate.

Finally, we examine the impact of temperature. Due to more rapid rundown at higher temperatures, most of our recordings were performed at room temperature (~22°C). Elevating the bath temperature also significantly shortened release intervals, from 2.3 ± 0.64 s (N = 16) at room temperature to 1.13 ± 0.67 s (n = 5) at 35°C (p<0.0001, unpaired t-test). While rundown was more rapid at 35°C, if we simply assume a similar amount of rundown as seen at room temperature, this suggests interevent intervals at 35°C as short as 615 ms (i.e., a frequency of 1.6 Hz). Alternatively, if we assume that the most rapid rates measured at 35°C come from the healthiest cells and are thus more representative of genuine rates in vivo, then interevent intervals one standard deviation below the mean of ~465 ms suggest a rate of 2.2 Hz. These various lines of reasoning converge on a

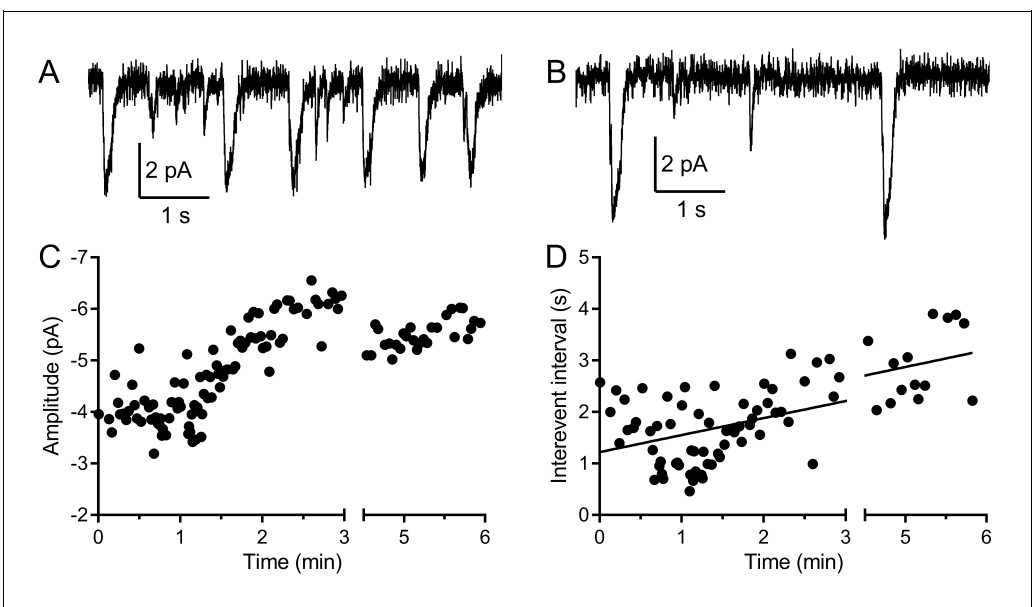

**Figure 3.** Multivesicular events increased in amplitude and declined in frequency during recording. (A, B) Example traces recorded from the same rod held at −40 mV during the first minute (A) and after 5 min (B). (C) The amplitude of multivesicular events as a function of time during the recording. (D) The intervals between multivesicular events as a function of time (room temperature). The plot of intervals (s) as a function of time after patch rupture (min) was fit with a linear regression (y-intercept = 1.22 s, slope = 0.33 s/min, R² = 0.42, n = 123 intervals).

spontaneous rate of multivesicular release events in darkness at physiological temperatures from intact mouse rods of ~2 Hz.

## Multivesicular release is sensitive to small voltage changes

We examined the impact of small voltage changes on the probability of multivesicular release events from rods. We hypothesized that multivesicular release may play a role in transmitting single-photon responses from rods because multivesicular release occurred most prominently in rods near the typical membrane potential in darkness and was not observed in cones. In mouse rods, the voltage changes evoked by single-photon absorption average ~3.5 mV (*Cangiano et al., 2012*). To test the sensitivity of multivesicular release to small voltage changes, we voltage-clamped the rod at –40 mV for 30 s and then applied a hyperpolarizing step to −43.5 mV. This small hyperpolarization consistently produced an appreciable pause in release (*Figure 4A*). The average interevent intervals measured over a 30 s period at −43.5 mV (3744 ± 2243, 129 intervals) were significantly longer than those measured in the same cells at −40 mV (2405 ± 1430 ms, 279 intervals, p<0.0001, unpaired t-test, n = 9 rods). In addition, the interval to the first multivesicular event observed after the step to

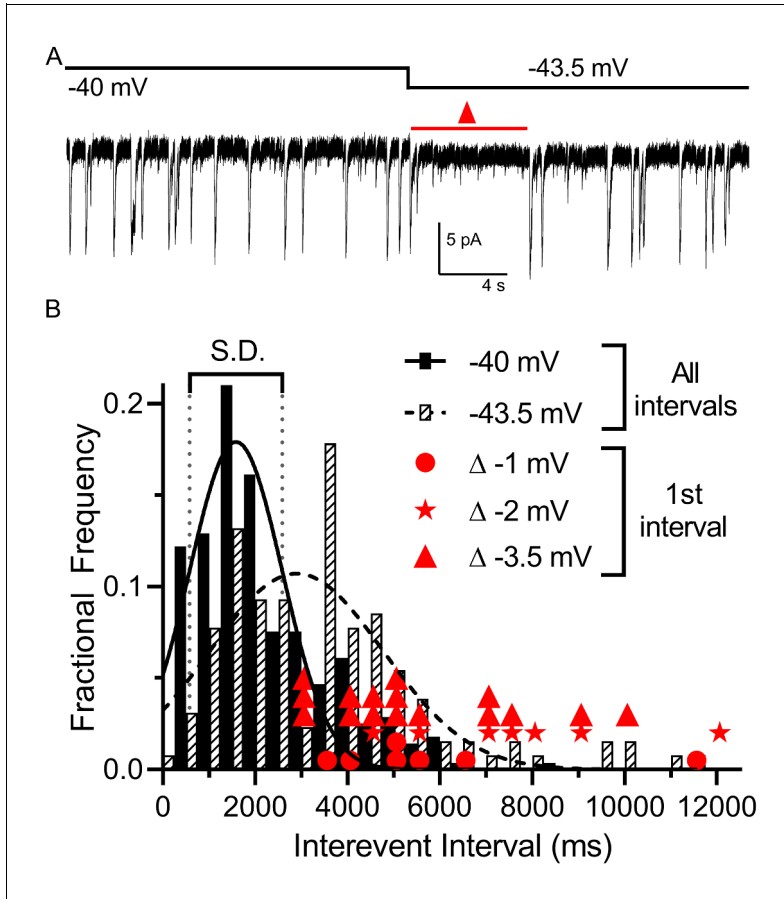

**Figure 4.** Multivesicular release was sensitive to small hyperpolarizing steps. (**A**) Example trace showing that multivesicular $I_{A(glu)}$ release events paused for several seconds immediately after hyperpolarizing the rod from −40 to −43.5 mV and the overall rate of these events remained slower for the remainder of the step. Red triangle denotes the time interval measured and plotted as red triangles in panel B. (**B**) Frequency distribution of intervals between multivesicular events measured at −40 and −43.5 mV (n = 9 rods), each fit with a single Gaussian (−40 mV, 1583 ± 1004 ms; −43.5 mV, 2896 ± 1872 ms). Hyperpolarization from −40 (279 events) to −43.5 (129 events) significantly extended the mean interval between multivesicular events (p<0.0001, unpaired t-test). Filled circles, stars and triangles show the intervals measured in individual trials immediately after hyperpolarizing steps of 1 (n = 7), 2 (n = 7), or 3.5 mV (n = 16) from a prior holding potential of −40 mV, respectively. All of these intervals were >1 standard deviation (vertical dotted lines) above the mean interval between multivesicular events measured at −40 mV.

−43.5 mV (red triangles in *Figure 4B* show measurements from individual trials) was consistently greater than 1 standard deviation above the mean interevent intervals measured at −40 mV. Single-photon responses of primate rods appear be smaller than those of mouse rods, averaging only ~1 mV (*Schneeweis and Schnapf, 1995*). We found that steps of −1 (n = 7; filled circles in *Figure 4B*) and −2 mV (n = 7; stars in *Figure 4B*) also produced appreciable pauses in multivesicular events immediately after the step. This lengthening of release intervals could potentially improve the signaling of single-photon responses to post-synaptic bipolar cells, allowing genuine reductions in release caused by a photon to be distinguished from a stochastic fluctuation in release.

To further investigate the capability of a single-photon absorption event to reduce multivesicular release, we clamped rods at −40 mV and applied a voltage waveform that mimicked the amplitude and kinetics of a single-photon voltage response in a mouse rod (*Figure 5*). This voltage waveform was constructed from a two exponential function that attained a peak amplitude of 3.4 mV after 245 ms and then declined with a time constant of 2.5 s (*Figure 5A*; *Cangiano et al., 2012*). In rods voltage-clamped at −40 mV, we repeated this single-photon voltage stimulus for 20–60 trials (*Figure 5A*). Binning the multivesicular release events in time revealed that release probability declined almost to zero at the peak of the simulated light response, falling in parallel with membrane hyperpolarization (*Figure 5B*; n = 1892 events from 28 rods). Multivesicular release then rebounded with an overshoot as the membrane potential recovered, showing ~50% more events initiated during the first few hundred milliseconds after termination of the simulated light response (between 3000 and 4000 ms, *Figure 5B*). The voltage stimulus did not truncate multivesicular events that had

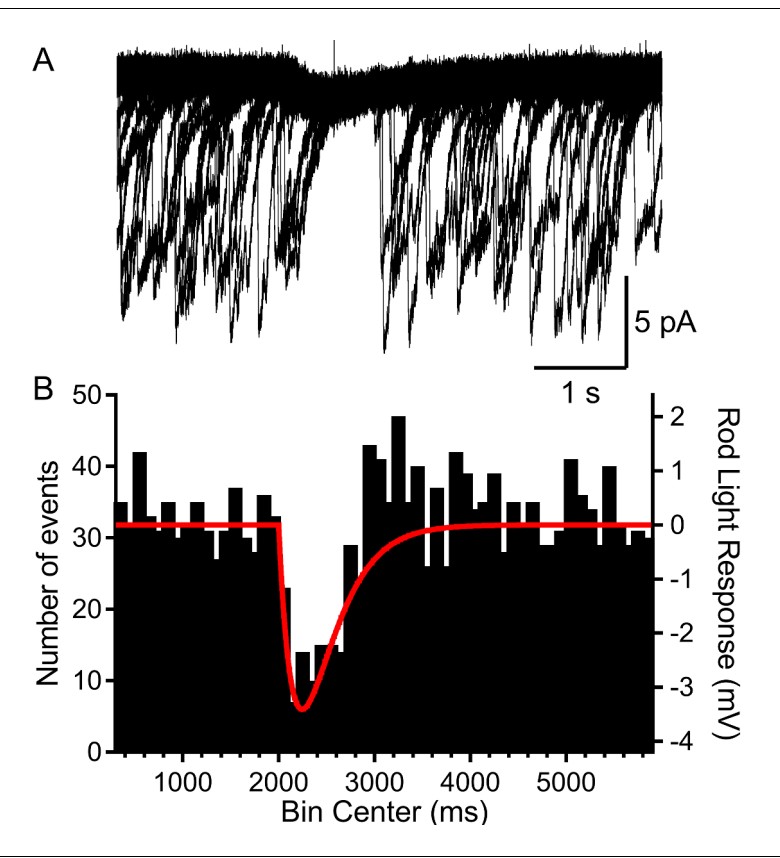

**Figure 5.** Rods voltage-clamped at −40 were presented a voltage stimulus that mimicked the single-photon response of mouse rods. (**A**) Example of a rod presented with the voltage stimulus 30 consecutive times, showing an appreciable pause in release during the single-photon voltage waveform. (**B**) Summary data showing the number of multivesicular events (1892 events, 28 rods) initiated before, during, and after the voltage stimulus (red trace). Multivesicular release events were almost never initiated during the single-photon waveform.

The online version of this article includes the following source data for figure 5:

**Source data 1.** Start times for multivesicular events during simulated single-photon light responses.

started prior to the stimulus; events that continued during the stimulus had the same number of quanta, measured by charge transfer, as prior events (p=0.53, t-test, n = 5 rods, not shown).

## White noise analysis

To identify voltage changes that preferentially evoke multivesicular release in rods, we held rods at −40 mV and applied a white noise voltage stimulus with a standard deviation of 3.5 mV and high frequency cutoff of 5 Hz, consistent with the rod power spectrum (*Chichilnisky and Rieke, 2005*). The average stimulus waveform that preceded each multivesicular event, that is, the event triggered average (ETA) waveform, consisted of a small hyperpolarizing excursion followed by a larger depolarizing excursion with a time course roughly similar to termination of a single-photon response (n = 4 rods; *Figure 6A*). Using a white noise voltage stimulus with a smaller standard deviation (0.8 mV), the ETA showed a waveform with a similar shape and time course (*Figure 6B*). When we tested a similarly small stimulus but with 20 Hz cutoff, the ETA appeared as noise (*Figure 6C*), suggesting the rod terminal preferentially responds to slow voltage changes with frequency response characteristics of rods in light and dark, filtering out higher frequency membrane voltage noise. These data reinforce the idea that transmitter release from rods is very sensitive to small, slow voltage changes similar to those experienced during a single-photon response, consistent with a role of these multivesicular events in transmitting small light responses at the rod ribbon synapse.

## Simulation of vesicle release dynamics indicates efficient transmission of the single-photon response

The previous experiments suggest the multivesicular release dynamics observed in rods held at −40 mV could be used for transmitting single-photon responses to rod-bipolar cells. We examined the consequences of these release statistics on transmitting single-photon responses by generating a model of rod release dynamics matched to those we measured and by utilizing an ideal observer analysis to examine the resulting signal fidelity (*Chichilnisky and Rieke, 2005*; *Field et al., 2019*; *Smith and Dhingra, 2009*).

Multivesicular release was modeled as a time-dependent Erlang process across a population of simulated rods forming the receptive field of a rod-bipolar cell (*Figure 7A*; Erlang release model,

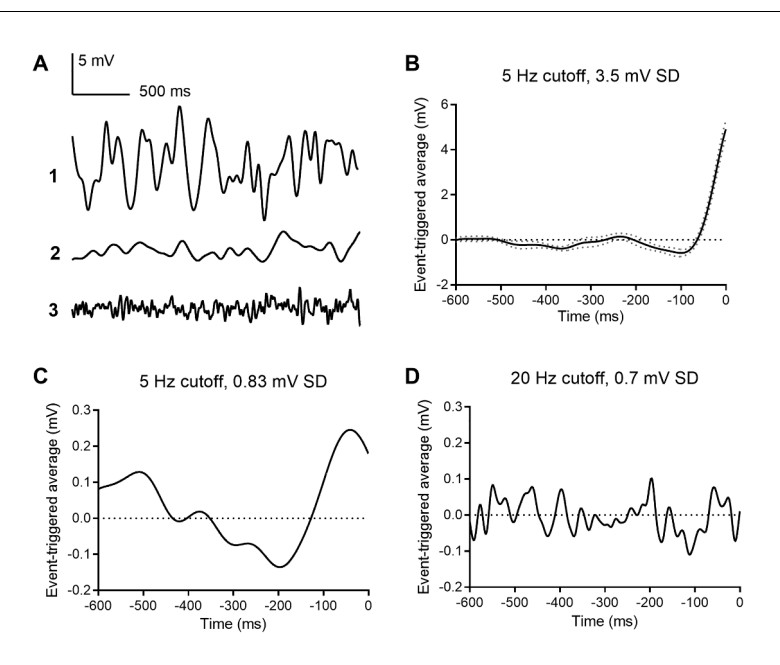

**Figure 6.** Event triggered averages (ETAs) compiled from responses to white noise voltage stimuli. (**A**) 2.5 s segments of white noise voltage stimulus with 5 Hz cut-off and 3.5 mV SD. (**B**) ETA evoked by the same white noise stimulus. (**C**) ETA evoked by a smaller stimulus with 5 Hz cutoff but only 0.8 mV SD. (**D**) ETA evoked by a white noise stimulus with 20 Hz cut-off and 0.7 mV SD.

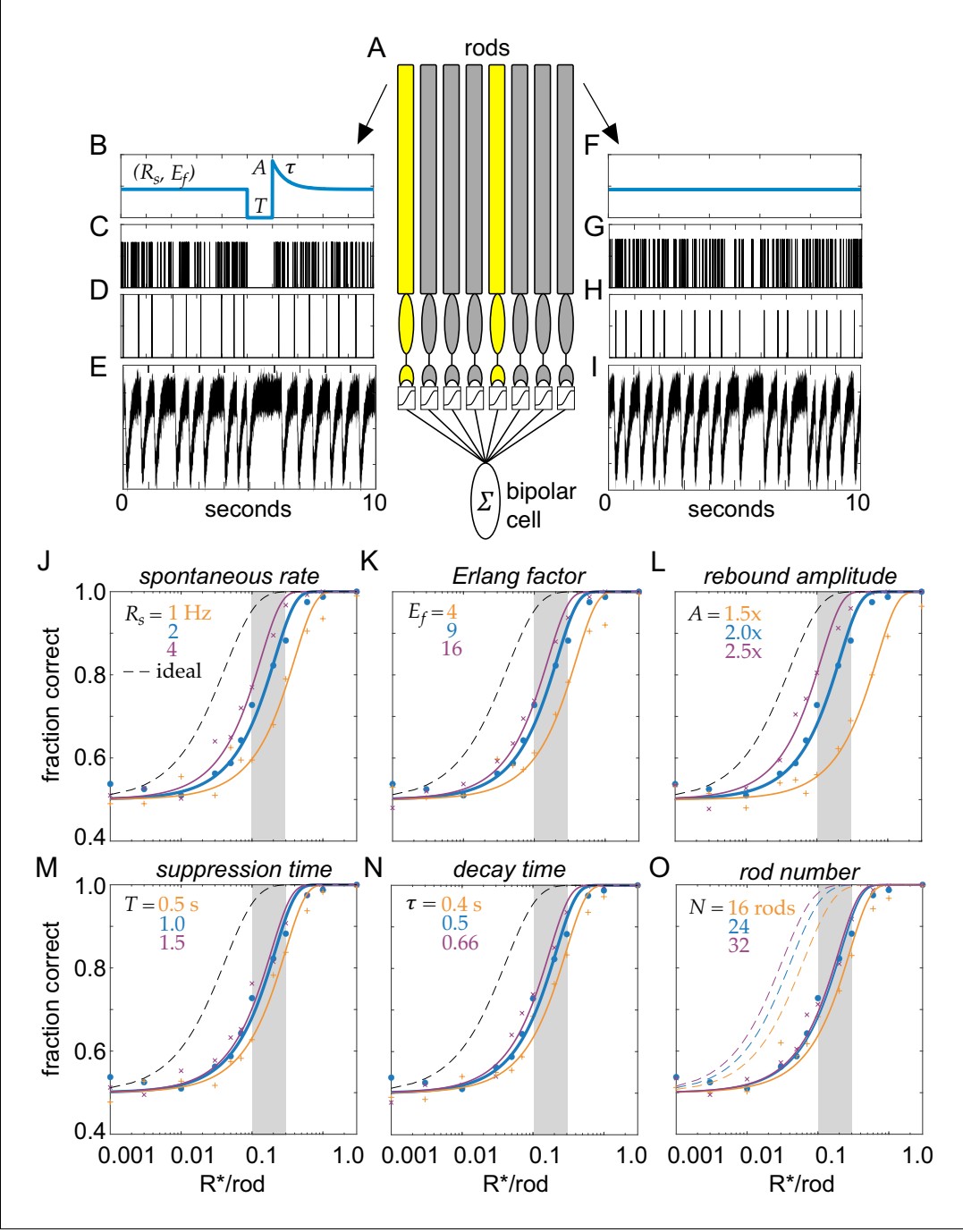

**Figure 7.** Model of clockwork multivesicular release can account for the detection threshold of rod-bipolar cells. (A) Schematic of a pool of simulated rod photoreceptors providing input to a simulated rod-bipolar cell. Yellow rods indicate cells absorbing a photon; gray rods indicate cells generating only noise. (B) Temporal dynamics of transmitter release illustrating the key parameters of the Erlang release model (see text for details). A flash is delivered 5 s into the trial. (C) Poisson counts underlying the Erlang process. (D) Accumulation of nine Poisson events in (C) triggers an Erlang event. (E) Erlang events in (D) convolved with function mimicking multivesicular release (*Equation 1*), with additive Gaussian noise. (F–I) Same as (B–E), but for a rod that does not absorb a photon. (J–O) Fraction correct in two-alternative forced-choice detection task as a function of flash strength. Points show output of model, and curves show cumulative Gaussian fits. Blue shows performance of model for parameters estimated from data. Orange and purple show performance for lower and higher model parameters, respectively, for spontaneous release rate (O), Erlang factor (R), rebound amplitude (L), suppression time (M), decay time (N), and rod number in pool (O).

see Materials and methods). In darkness, the release probability was constant in time (*Figure 7F*); the absorption of a photon caused a suppression of release lasting T seconds (*Figure 7B*). Note, this will generate the same integrated change in release probability as exponential decay with a time constant of T. Following suppression, release probability rebounded to an amplitude of A above baseline and then decayed exponentially back to baseline with a time constant $\tau$ (*Figure 7B*). To generate Erlang release statistics, we first sampled an inhomogeneous Poisson process (*Figure 7C*). Poisson events were counted until reaching the Erlang factor, $E_f$. When the counter reached $E_f$, this generated a release event (*Figure 7D*) and the counter was reset. This yielded sub-Poisson variability with a coefficient of variation less than a Poisson process by 1/sqrt($E_f$). Release events were convolved with a template, and independent Gaussian noise was added to approximate recorded rod release events (compare *Figure 7E* with *Figure 2B*).

The fidelity of rod output with these vesicle release dynamics was assayed using a two-alternative force-choice (2AFC) task and an ideal observer analysis (*Chichilnisky and Rieke, 2005*; *Field et al., 2019*; *Smith and Dhingra, 2009*). For the 2AFC task, one response was generated by simulating a flash of strength S delivered to a population of N rods. A second response was generated with no flash. The task of the ideal observer was to identify the response containing the flash. The ideal observer was constructed by calculating the optimal linear discriminant separating flash from no-flash trials in each individual rod. The dot product between this template, and each individual response was calculated and then weighted by the likelihood it was generated by the given flash strength (see Materials and methods). This procedure instantiates an optimal linear filtering of the rod output followed by an optimal non-linear weighting that approximates the non-linear threshold performed at the rod-to-rod-bipolar synapse (*Field and Rieke, 2002b*; *Field et al., 2019*). The fraction of trials in which the flash response was correctly identified was calculated over 400 trials at each flash strength; flash strengths ranged from 0.001 to 1.0 photo-isomerizations (Rh*) per rod.

For a simulated population of 24 rods, the probability of detecting flashes varied smoothly with flash strength, reaching 75% correct in the 2AFC task between 0.1 and 0.2 Rh*/rod (*Figure 7J*, blue curve). This performance fell short of an optimal photon detector that is limited only by Poisson variability in the number of photons absorbed from trial-to-trial (*Figure 7J*, dashed curve) (*Field et al., 2019*). However, flash detection performance compared well to previous measurements of rod-bipolar cell sensitivity, which place their detection threshold (75% correct in the 2AFC task) between 0.1 and 0.3 Rh*/rod (*Figure 7J*, vertical gray bar) (*Field and Rieke, 2002b*; *Okawa et al., 2010*). Thus, the performance of the simulated rod output with temporally regular multivesicular release falls within the range of flash sensitivities demanded by rod-bipolar cell measurements. This indicates that the observed rod vesicle release dynamics can account for rod-bipolar cell flash sensitivity.

We next examined how this result depended on different choices of model parameters: while the chosen parameters were taken from our measurement of rod vesicle release (biased toward the earliest portion of the recordings and the highest temperatures), our recordings allow for a range of potential values under physiological conditions. We examined how detection performance depended on the multivesicular event rate ($R_s$), the Erlang factor ($E_f$), the rebound amplitude (A), the suppression time (T), the decay time ($\tau$), and the number of rods in the pool (N) (*Figure 7J–O*). In general, the detection threshold was not strongly impacted over a reasonable range of these model parameters: the flash strength generating 75% correct performance was consistent with the detection threshold of rod-bipolar cells (*Figure 7J–O*, gray bars).

Two features of this multivesicular release model distinguish it from previous examinations of rod vesicle release. The first feature is multivesicular release. Each multivesicular event contains ~17 vesicles. With a release rate of ~2 multivesicular events/s, this means the rod terminal will release ~34 vesicles/s. Thus, we analyzed how performance of the multivesicular release model compared with asynchronous Poisson release of 34 vesicles/s (*Figure 8A*; Poisson release model, see Materials and methods). For comparison, we also included performance of an ideal detector and that predicted from the rod photocurrent (*Figure 8A*; rod photocurrent model, see Materials and methods). The temporally regular multivesicular release model exhibited higher sensitivity than asynchronous Poisson release, given a fixed vesicle budget of 34/s. Thus, the observed statistics of vesicle release from the rod terminal appear to be efficient relative to asynchronous release, given a fixed vesicle budget.

The second feature distinguishing this model is the temporal regularity of vesicle release. The idea of temporally regular release has been postulated previously (*Schein and Ahmad, 2005*), but

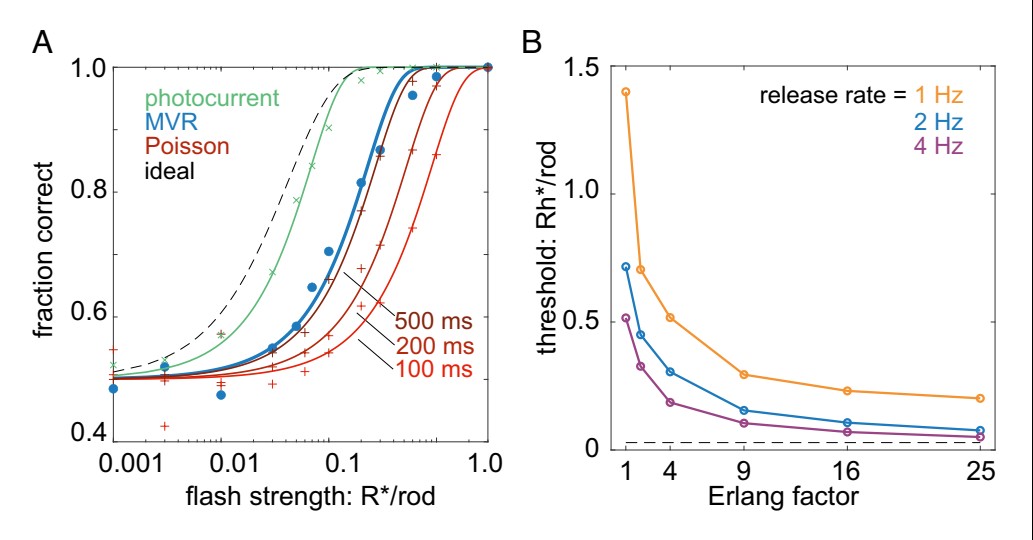

**Figure 8.** Temporally regular multivesicular release supports flash detection. (**A**) Comparison of different models in 2AFC detection task. Black dashed curve is performance of an ideal photon detector. Green is performance predicted from rod photocurrents. Blue is performance of the multivesicular Erlang release model (*Figure 7*). Red curves show performance of asynchronous Poisson release for three different release suppression durations (100, 200, and 500 ms). Points show simulation results; curves show cumulative Gaussian fits. (**B**) Comparison of detection thresholds (75% correct performance) in the 2AFC task for different release rates and Erlang factors, using the multivesicular release model.

not shown experimentally or considered in the context of multivesicular release. The temporal regularity of release is modeled here by the Erlang factor ($E_f$), which reduces variability in the intervals between multivesicular events. Our data showed a reduction in interevent variability relative to a Poisson process, consistent with Erlang factors of 3–12 (average 6; see *Figure 2*). Examining detection thresholds as a function of the Erlang factor revealed that over a range of multivesicular release rates, detection thresholds fell considerably as the Erlang factor increased to four and Erlang factors greater than nine exhibited rapidly diminishing returns on detection thresholds (*Figure 8B*). Thus, further increasing the regularity of interevent intervals (increasing the Erlang factor above 9) yields relatively small improvements to the fidelity of transmitting single-photon responses to rod-bipolar cells.

## Discussion

To better understand how responses of rods to single photons can be transmitted reliably to downstream neurons, we examined the mechanisms by which ongoing release from rods can be regulated by small changes in membrane potential. Using $I_{A(glu)}$ as a presynaptic assay for release, we found that at a membrane potential similar to that in darkness, rods released glutamate-filled vesicles in coordinated multivesicular release events at rates that were not well modeled by Poisson statistics. Multivesicular events occurred at more regular intervals consistent with an Erlang distribution in which events occurred after waiting 3–12 Poisson intervals. Multivesicular release events were also extremely sensitive to small voltage changes. Simulating the impact of these properties on release showed that they can improve the ability of rod-bipolar cells to discriminate suppressed release caused by absorption of a single photon from stochastic pauses in release.

The ability of the vertebrate visual system to detect single photons arises from high photon capture rates combined with high gain in the phototransduction cascade of rods. Next, the relatively small changes in membrane voltage produced by absorption of a single photon must produce an appreciable change in the rate of release at the rod synapse. At the synapse from rods to rod-bipolar cell dendrites, there is a non-linear thresholding mechanism in which only responses that exceed a certain size are transmitted, removing baseline noise along with small responses that fall below

that threshold (**Berntson et al., 2004**; **Field and Rieke, 2002b**; **Sampath and Rieke, 2004**). In mouse retina, where 20–35 rods converge onto each rod-bipolar cell (**Behrens et al., 2016**; **Calkins and Sterling, 1999**; **Rao-Mirotznik et al., 1998**; **Tsukamoto and Omi, 2013**), comparisons between rods and rod-bipolar cells showed thresholds that excluded 40–85% of the single-photon events in rods (**Berntson et al., 2004**; **Field and Rieke, 2002b**; **Schein and Ahmad, 2006**). In rabbit retina, where 100 rods may converge onto each rod-bipolar cell, as many as 90% of the single-photon responses are removed by the thresholding mechanism (**Trexler et al., 2011**). Finally, as many as 100,000 rods eventually converge onto a single ganglion cell, allowing the capture of scarce single photons to produce detectable changes in retinal output (**Field et al., 2019**; **Goodchild et al., 1996**; **Okawa and Sampath, 2007**; **Takeshita et al., 2017**; **Taylor and Smith, 2004**).

The modeling presented here (**Figure 7**) does not strongly constrain the fraction of single-photon responses that fail to traverse the synapse with rod-bipolar cells. This is because these simulations did not incorporate variability in the voltage of the single-photon voltage response or fluctuations in the membrane potential due to continuous noise in phototransduction. Instead, the model began with the estimated dynamics of transmitter release during a simulated light flash (**Figure 7B**), bypassing these upstream noise sources. To understand how this mode of transmitter release impacts the optimal location of the non-linear threshold at the rod to rod-bipolar synapse would require incorporating noise in phototransduction and determining how this noise (both single-photon response variability and continuous noise) influences the observed temporal variability in multivesicular release events.

The non-linear thresholding mechanism used at rod-bipolar synapses is thought to occur before summation of rod responses in the rod-bipolar cell soma (**Field and Rieke, 2002b**; **Sampath and Rieke, 2004**). A key requirement for this non-linear thresholding is that glutamate levels in the cleft must be maintained at levels high enough to keep the rod-bipolar cell glutamate receptors saturated. By relieving saturation in the receptor signaling cascade, larger reductions in glutamate release have disproportionately larger effects (**Sampath and Rieke, 2004**). In addition to improving regularity, multivesicular release by rods may help ensure that glutamate levels remain high enough at individual synapses to maintain saturation in darkness.

AII amacrine cells show synchronized membrane currents in darkness that arise from synchronous rod inputs into rod-bipolar cells (**Grimes et al., 2014**). Two rod-bipolar cell dendrites terminate beneath each rod ribbon in mouse retina (**Behrens et al., 2016**). Together with electrical coupling among rods via cones (**Jin et al., 2020**), shared responses of two such bipolar cells evoked by large multivesicular release events might contribute to the synchrony among AII amacrine cells.

Our previous recordings of release from a sample of mouse rods suggested an overall rate of release at −40 mV of ~11 v/s at 35°C (**Hays et al., 2020b**). However, assessments of rundown in the present study suggest that the true rate in vivo prior to patch rupture may be twice this average rate. Earlier computer simulations suggested that, assuming a purely Poisson release process, postsynaptic detection by rod-bipolar cells would require rod release rates of 80–100 v/s (**Rao et al., 1994**; **van Rossum and Smith, 1998**). Consistent with this conclusion, our model indicated that Poisson release at a rate of 34 v/s would not provide the level of performance observed in rod-bipolar cells unless we employed an unusually long suppression duration of 500 ms.

A second strategy suggested to overcome noise in release rates is to make release more regular. At hyperpolarized membrane potentials, we found that glutamate release from rods followed a Poisson distribution similar to that seen at many other neurons (**Malagon et al., 2016**; **Miki, 2019**). This nearly ubiquitous feature of neurons was established in the 1950s by Bernard Katz who modeled spontaneous release events to develop the quantal hypothesis of neurotransmission (**Kavalali, 2015**; **Kavalali, 2018**). The occasional unitary events observed in rods held at −40 mV also showed a Poisson distribution of intervals. However, the dominant feature of release at −40 mV was the appearance of multivesicular release events occurring at fairly regular intervals. The distribution of intervals between multivesicular events could not be explained by Poisson statistics and were instead more consistent with an Erlang process, having a CV narrower than expected from a Poisson process. By waiting a specific number of Poisson intervals between each Erlang event, Erlang processes effectively average the underlying Poisson intervals and thereby reduce variability. It was suggested previously that an Erlang process might play a role in transmitting single-photon responses from rods (**Schein and Ahmad, 2005**; **Schein and Ahmad, 2006**). Our experimental data showed a narrower distribution than expected for a Poisson process consistent with the accumulation of as many as 12

Poisson events before each multivesicular release event. The cumulative wait times for replenishment of multiple vesicles to the releasable pool at the base of the synaptic ribbon provides a possible mechanism for achieving an Erlang-like process to enhance regularity in release rates. Predictions of Schein and Ahmad suggested that achieving reliable single-photon detection with a release rate of 100 v/s/ribbon would require an Erlang process with an Erlang factor of 66 (i.e., an accumulation of 66 Poisson intervals) (*Schein and Ahmad, 2006*). Our model, which among other things differed by including a thresholding non-linearity at the rod-to-rod bipolar cell synapse, showed that even a modest increase in regularity combined with multivesicular release can improve the post-synaptic detection of single-photon events.

Another surprising characteristic of the multivesicular release in rods was its sensitivity to small voltage changes. While the overall rate of release parallels changes in $I_{Ca}$, a small hyperpolarizing voltage change caused appreciable pauses in release. The average interevent interval dropped close to zero during a simulated single-photon response, rather than declining only ~20–25% as predicted from the decline in $I_{Ca}$. Hyperpolarizing the membrane by only 1 mV, similar to the amplitude of single-photon responses in primate rods (*Schneeweis and Schnapf, 1995*), also caused an appreciable pause in release. The mechanisms that coordinate release among vesicles to produce this exquisite sensitivity remain unclear.

Recent studies on mice in which synaptic ribbons have been eliminated by deleting the key ribbon protein, Ribeye, showed surprisingly modest deficits (*Fairless et al., 2020*; *Okawa et al., 2019*) Multivesicular release in both salamander and mouse rods involves ribbon-associated vesicles (*Hays et al., 2020a*; *Hays et al., 2020b*). One role of the ribbon may thus be to help organize vesicles to promote multivesicular release, implementing an Erlang process to improve the encoding of light responses near scotopic threshold (*Schein and Ahmad, 2005*).

The need to rapidly recycle vesicles is a barrier to rods maintaining high release rates. Rod spherules in mouse retina possess 5800–7500 vesicles apiece (*Zampighi et al., 2011*) and so the entire pool of vesicles would be depleted in roughly a minute if release were maintained at 100 vesicles/s. Our results suggest that the temporally regular multivesicular release strategy employed by rods provides an efficient means of encoding small voltage changes to meet both constraints on high release rates and requirements for reliable signaling at rod ribbon synapses (*Sterling and Laughlin, 2015*).

## Materials and methods

### Animals

Control C57/Bl6 mice (RRID:IMSR_JAX:000664) were kept on 12 hr dark–light cycles, and animal handling protocols were approved by the University of Nebraska Medical Center Institutional Animal Care and Use Committee. Mice of both sexes aged 4–12 weeks were euthanized in accordance with AVMA Guidelines for the Euthanasia of Animals by $CO_2$ asphyxiation followed by cervical dislocation (protocol 18-077-06).

### Electrophysiology

Recordings from rods were performed using a flatmount preparation of isolated retina. Immediately after euthanizing the mouse, it was enucleated and eyes were placed in Ames' medium (US Biological; RRID:SCR_013653) bubbled with 95% $O_2$/5% $CO_2$. The cornea of an eye was punctured with a scalpel and the anterior segment removed. The retina was isolated by cutting optic nerve attachments. After making four fine cuts at opposite poles of the retina, it was flattened photoreceptors face up onto a glass slide in the perfusion chamber. The retina was anchored in place using a brain slice harp (Warner Instruments, cat. no. 64–0250). The perfusion chamber was placed on an upright fixed-stage microscope (Nikon E600FN) with a $60\times$ water immersion, long-working distance objective (1.0 NA). Unless otherwise noted, the retina was superfused with room temperature Ames' solution bubbled with 95% $O_2$/5% $CO_2$ at ~1 mL/min. Prior to recording, outer segments from a region of the retina were removed by gentle suction applied through a broken patch pipette. Rod cell bodies were identified morphologically and targeted with positive pressure using recording electrodes mounted on Huxley–Wall micromanipulators (Sutter Instruments).

Rod recordings were performed in whole-cell voltage-clamp using a Multiclamp 700A amplifier (Axon Instruments/Molecular Devices) and signals digitized with a DigiData 1550 (Axon Instruments/Molecular Devices). Data acquisition and analysis was performed using pClamp 10 Software (Molecular Devices; RRID:SCR_011323). Voltages were not corrected for liquid junction potential (KSCN pipette solution: 3.9 mV). Most experiments were performed in room light, but in some experiments, retinas were prepared and electrodes positioned in darkness under infrared illumination using Gen III night vision goggles (Nitemate NavIII, Litton Industries).

Intracellular pipette solutions for $I_{A(glu)}$ measurements contained (in mM): 120 KSCN, 10 TEA-Cl, 10 HEPES, 1 CaCl$_2$, 1 MgCl$_2$, 0.5 Na-GTP, 5 Mg-ATP, 5 EGTA, 5 phosphocreatine, pH 7.3. Rods were distinguished from cones by their smaller membrane capacitance. Passive membrane properties of rods averaged $C_m = 3.2 \pm 0.2$ pF; $R_m = 2.3 \pm 0.04$ GΩ (mean ± SD, n = 20).

$I_{A(glu)}$ event frequency, kinetics, and charge transfer were measured from events identified with the event finder function in pClamp directed by a template that included a sampling of ~10 manually curated events.

## Rod vesicular release and photocurrent models

### Erlang release model (model 1)

Multivesicular release from the rod terminal was modeled as an Erlang process (*Figure 7*). To generate stochastic Erlang events with a spontaneous event rate of $R_s$ and a coefficient of variation $1/\sqrt{[2]}E_f$, a Poisson process was first generated with an event rate of $R_s * E_f$. For every $E_f$ Poisson events that were generated, an Erlang event was generated producing stochastic events that were $\sqrt{[2]}E_f$ more regularly spaced in time than a Poisson process (see *Figure 7B–D and F–H*). These discrete events were modeled as delta-functions and then convolved with an alpha waveform:

$$M(t) = a * \frac{t}{\tau_r} * e^{\frac{1-t}{\tau_r}} \tag{1}$$

Finally, independent Gaussian noise was added (*Figure 7E,I*). To model the release dynamics following a single-photon response, the Poisson process was made time dependent (*Figure 7B*), such that following the absorption of a simulated photon, the Poisson rate was set to zero for a $T$ seconds, then set to $A$ and exponentially relaxed back to the baseline rate with a time constant of $\tau$. In these simulations, time was sampled at 0.1 ms to ensure the probability that the Poisson process generated a value other than 0 or 1 was miniscule (<0.0001). In the unlikely event that a value > 1 was sampled, the value was set to 1.

### Poisson release model (model 2)

To compare the flash detection performance of multivesicular release governed by an Erlang process to that of asynchronous Poisson release (*Figure 8*), a similar procedure was followed as that described in the preceding paragraph. However, Poisson events were not accumulated to generate an Erlang event and the suppression of the vesicular release following a simulated single-photon response was set to 25% of the spontaneous release rate to match previous estimates given the membrane potential of rods and given the voltage dependence of calcium channel activation (*Schein and Ahmad, 2005*; *Schein and Ahmad, 2006*).

### Rod photocurrent model (model 3)

To model the rod photocurrent and the detection sensitivity limits set by photocurrent noise sources (*Figure 8*), we used a generative model of the rod response (*Field et al., 2019*). A rod response, $r(t)$, was generated from the following equation:

$$r(t) = \sum_{n=1}^{N} \left( r_\mu(t) + \sum_i w_i c_i(t) \right) + \sum_{d=1}^{N_d} \left( r_\mu(t - t_d) + \sum_j w_j c_j(t - t_d) \right) + \eta(t) \tag{2}$$

Continuous dark noise, $\eta(t)$, was generated by sampling from a Gaussian distribution and filtered in time to match the power spectrum of measured continuous noise (*Field et al., 2019*). $N$ photon responses in an individual rod were generated by the first term in *Equation 2* by sampling from a Poisson distribution with a mean given by the flash strength on a given trial. The mean single-photon

response is given by $r_\mu(t)$, and the covariance of the single-photon response is captured by summing over a weighted set, $w_i$, of eigenvectors, $c_i(t)$, derived from the covariance matrix of the single-photon response (*Field et al., 2019*). Finally, discrete noise events caused by the thermal activation of rhodopsin were captured by the second term of *Equation 2*: $N_d$ thermal isomerizations were generated using the same formulation as for the single-photon response, but $N_d$ was determined by sampling from a Poisson distribution with a mean given by the thermal isomerization rate at 35°C and each isomerization event occurred at random and independent times given by $t_d$.

In the original formulation of this model (*Field et al., 2019*), the noise parameters were derived from measurements of the photocurrent from primate rods. We adjusted these parameters to account for the greater amounts of continuous noise and variability in the single-photon response present in mouse rods (both relative to the amplitude of the mean single-photon response). Specifically, the continuous noise was increased by 22% and single-photon response variability was increased by 37.5% (*Field and Rieke, 2002a*; *Field and Rieke, 2002b*). The kinetics of the single-photon response and the shape of the power spectrum of the continuous noise are both similar between mouse and primate rods, so no adjustments were made to these quantities. The thermal rate in mammalian and mouse rods is somewhat uncertain, with the literature allowing for a wide range of values between 0.001 and 0.015 Rh*/rod/s, so we chose an intermediate value of 0.005 Rh*/rod/s (*Burns et al., 2002*; *Field et al., 2019*; *Fu et al., 2008*; *Yue et al., 2017*). Importantly, this model for simulating rod responses reproduces rod detection and temporal sensitivities in a 2AFC task (*Field et al., 2019*), supporting the application of this model for analyzing task performance of pools of rods.

## 2AFC task and ideal observer analysis

Using each of the three models described in the previous section, detection threshold was determined using a 2AFC task and an ideal observer analysis. For each pair of trials, the task was to determine which contained the 'flash' and which did not. The ideal observer was constructed from a training set of simulated flash and no-flash (null) responses (1000 of each). Note, in the training phase, the simulated flash responses consisted only of single-photon responses without Poisson fluctuations in the number of photons absorbed. From these simulated responses, a difference-of-means linear discriminant was constructed (*Duda et al., 2001*):

$$D = \mu_{flash} - \mu_{null} \tag{3}$$

where D is the linear discriminant, $\mu_{flash}$ and $\mu_{null}$ are the mean of the flash and no-flash responses, respectively. Then the dot products of the flash and no-flash responses with the discriminant were calculated; each set of projections yielded an approximately Gaussian distribution associated with the flash and no-flash trials. Thus, their means and standard deviation were used to summarize the distributions: $\mu_A$, $\mu_B$, $\sigma_A$, and $\sigma_B$, where A and B denote the distributions of flash and no-flash responses, respectively, after computing their dot products with the discriminant. Then a new 'test' set of flash and no-flash responses were simulated and discriminated by the ideal observer using a Bayes-optimal non-linear weighting.

$$R = \sum_i^P w_i(r_i(t) \cdot D) \tag{4}$$

where P is the number of rods in the pool, r(t) is a given response (flash or no flash), D is the discriminant from *Equation 3*, and $w_i$ is the non-linear weight given by:

$$w_i = \frac{G(r_i(t) \cdot D | \mu_A, \sigma_A) \times P(1|f)}{G(r_i(t) \cdot D | \mu_B, \sigma_B) \times P(0|f)} \tag{5}$$

$G(X|\mu,\sigma)$ is the probability of sampling X from a Gaussian distribution with mean, $\mu$, and standard deviation $\sigma$; $P(Y|\lambda)$ is the probability of sampling Y from a Poisson distribution with mean $\lambda$; and f in *Equation 5* is the flash strength used to generate the 'flash' responses, which acts like a prior in the Bayesian sense. Note, in the 'test' set, the simulated flash responses included Poisson fluctuations in the number of photons absorbed from trial-to-trial.

Responses were simulated for flash strength *f* and compared to responses of flash strength 0. If $R_f$ *(flash response)* > $R_0$ (no-flash response), the trial was scored as correct; if $R_f < R_0$, the trial was scored as incorrect; and if $R_f = R_0$, the trial was randomly scored as correct with 50% probability.

## Statistical analysis

Statistical analysis was performed on GraphPad Prism versions 7–9 (RRID:SCR_002798), and all data are represented as mean ± SD unless otherwise noted. Recordings were performed using rods from at least three different retinas.

To test whether release rates reflected a Poisson process, interevent interval histograms were fit with a one-phase exponential decay. In a second approach, the number of release events was tallied over a long period (30 s) to calculate the average number of release events per second (λ). A fractional frequency distribution of events was then made using 1 s bins and fit with a Poisson model:

$$P(x) = \frac{\lambda^x e^{-\lambda}}{x!} \qquad (6)$$

where *P* is the probability of observing *x* number of events and *e* is Euler's constant.

## Acknowledgements

Funding was provided by UNMC Graduate Fellowship (CLH) and NIH grants EY10542 and EY32396 (WBT).

## Additional information

### Funding

| Funder | Grant reference number | Author |
|---|---|---|
| National Eye Institute | EY10542 | Wallace B Thoreson |
| National Eye Institute | EY32396 | Wallace B Thoreson |
| UNMC College of Medicine | Graduate Fellowship | Cassandra L Hays |

The funders had no role in study design, data collection and interpretation, or the decision to submit the work for publication.

### Author contributions

Cassandra L Hays, Conceptualization, Data curation, Formal analysis, Investigation, Methodology, Writing - original draft, Writing - review and editing; Asia L Sladek, Investigation, Writing - review and editing; Greg D Field, Conceptualization, Software, Formal analysis, Investigation, Methodology, Writing - original draft, Writing - review and editing; Wallace B Thoreson, Conceptualization, Data curation, Formal analysis, Supervision, Funding acquisition, Investigation, Methodology, Writing - original draft, Project administration, Writing - review and editing

### Author ORCIDs

Cassandra L Hays  https://orcid.org/0000-0003-4481-3495
Greg D Field  https://orcid.org/0000-0001-5942-2679
Wallace B Thoreson  https://orcid.org/0000-0001-7104-042X

### Ethics

Animal experimentation: Animal handling protocols were approved by the University of Nebraska Medical Center Institutional Animal Care and Use Committee. Mice of both sexes aged 4-12 weeks were euthanized in accordance with AVMA Guidelines for the Euthanasia of Animals by $CO2$ asphyxiation followed by cervical dislocation (protocol 18-077-06).

Decision letter and Author response
Decision letter https://doi.org/10.7554/eLife.67446.sa1
Author response https://doi.org/10.7554/eLife.67446.sa2

## Additional files

### Supplementary files

• Transparent reporting form

### Data availability

All data generated or analyzed during this study are included in the manuscript and supporting file. Source data for Figure 5B are provided.

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
