## [Decision Letter]

Thank you for submitting your article "Properties of multi-vesicular release from mouse rod photoreceptors support transmission of single photon responses" for consideration by *eLife*. Your article has been reviewed by 3 peer reviewers and the evaluation has been overseen by a Reviewing Editor and a Senior Editor.

All three reviewers were extremely enthusiastic about the quality of the data and the impact of the findings. You are strongly encouraged to look at the suggestions from all three reviewers for clarifications. There is one major suggestions made by Reviewer #3 that it would be great for you to implement. After discussion amongst reviewers, all three felt it was an important suggestion but they trusted you would do it and that they didn't need to see the revision. Please let me know if this additional work on the simulations cannot be done for some reason.

Reviewer #1:

Hays et al. studied synaptic transmission from rod photoreceptors in the mouse retina to address a fundamental question in visual neuroscience: how is the retina able to encode the absorption of a single photon? It is well known that rod photoreceptors can generate an electrical response to the absorption of a single photon and that, remarkably, the rod output can be transmitted reliably across multiple synapses in the retinal circuit to modulate action potential firing by retinal ganglion cells, the retinal output neurons; ultimately, the responses of individual ganglion cells to just a few photons absorbed by the entire population of retinal rods can generate visual percepts in human observers and drive visually-guided behavior in experimental animals. The ability of the retinal circuit to encode the absorption of single photons requires that it possess cellular and synaptic mechanisms to amplify very small neural signals and to reject noise generated by stochastic cellular and synaptic processes. Past work by Field and Rieke (2002) and Sampath and Rieke (2004) demonstrated a thresholding mechanism in the rod bipolar cells postsynaptic to the rods that rejects noise and enhances the reliability of signaling at the rod-rod bipolar cell synapse in the mouse retina. Here, Hays et al. show that multi-vesicular release occurring at a regular interval from the rod terminal provides a reliable signal to the postsynaptic thresholding mechanism to optimize signal detection.

The strength of this study is the combination of experiment and computation. Experimentally, the authors use electrophysiological recordings of glutamate transporter currents from single rods to demonstrate that large, multivesicular release events from rod presynaptic active zones occur at regular intervals described by a an Erlang distribution and, remarkably, that hyperpolarization of the rod by just about 3 mV-the hyperpolarization induced by the absorption of a single photon-causes a reliable and periodic cessation in multivesicular release. Computationally, the authors implement an ideal observer model to demonstrate that regular multivesicular release provides an optimal input to the rod bipolar cell.

There are no significant weaknesses to this study. One must acknowledge, as the authors do, the limitations of the electrophysiological recordings: namely, time-dependent changes in the rate and size of the transporter currents reflecting recording-induced changes in the presynaptic release process. Concern about this so-called run-down is mitigated by the authors' ability to modulate release using physiological stimuli (e.g. Figures 4-6); significant degradation of presynaptic active zone function would not permit this.

In summary, this is an absolutely fantastic study that provides what should be a final answer to a long-standing and vexing question: how is transmission at the rod-rod bipolar synapse optimized to maximize the signal-to-noise ratio in the encoding of retinal responses to single photons?

I have no recommendations: this is one of the best submitted manuscripts that I have reviewed in many years. I will note that a reference to Field et al. on line 458 is missing the year. I do have some mild concerns about the run-down of the transporter currents, but this effect is unavoidable and addressed by the authors. I do wonder how sensitive the multivesicular release process is to local and global changes in intracellular [Ca^2+^], but this is a topic for another study.

Reviewer #2:

In this study, the authors take advantage of glutamate anion currents (IA(glu)) on rod photoreceptors to monitor their neurotransmitter release kinetics. While rods can sense single photons, the synaptic mechanisms underlying this sensitivity has been poorly understood. The authors propose that while at low membrane potentials rods release glutamate tonically in a quantal manner, depolarization to near physiological potentials (~-40mV) and enhanced Ca^2+^ concentrations result in an increase in propensity of multivesicular release leading to emergence of a pulsatile form of release. Under these conditions, even mild hyperpolarizations elicit rapid suppression of multivesicular release and thus robustly reflect single photon responses in the neurotransmitter release output.

The authors should provide further clarification to the following issues prior to publication:

1. The authors indicate that anionic currents associated with presynaptic glutamate transport vary linearly with glutamate levels in the synaptic cleft. However, a close examination of the cited work (Hasegawa et al., 2006; Otis and Jahr, 1998) reveal several non-linear regions in the glutamate – IA(glu) dose response. Therefore, the authors should provide further evidence for linearity and ensure that their measurements (especially the large multivesicular events) are indeed within the linear region of detection.

2. In the manuscript, it is not clearly stated whether the resting Ca^2+^ independent release is independent of the multivesicular release seen at elevated Ca^2+^. In my reading, the data suggests that the baseline Poisson process persists even at higher membrane potentials while the multivesicular release process emerges independently. Or do the authors think the two forms of release (univesicular and multivesicular) have the same origin? Further clarification of this issue will be useful.

Reviewer #3:

It has long been known that dark-adapted humans can detect just a handful of photons in a small region of the retina. This finding requires low noise in synaptic transmission. This paper addresses the question on how single photons arising in rods can be transmitted with high fidelity through the first retinal synapse. There has been a major gap in our understanding on the precise nature of glutamate release dynamics in this synapse and on how this synapse allows the transmission of single-photon signals with such a high precision. This paper characterizes glutamate release statistics presynaptically by careful single-cell electrophysiological recordings and shows that high-fidelity encoding of single-photon responses arises from regularly timed multivesicular release events that lead to less variance in inter-event interval than a conventional Poisson process. This paper is the first one providing evidence that tightly controlled multi-vesicular glutamate release statistics in rod photoreceptors is the key for mediating single-photon responses. Thereby this manuscript is of high interest to a wide audience in neuroscience.

I have only one major comment for the authors.

There has been debate on the precise amount of single-photon responses eliminated by the nonlinear thresholding in between rods and rod bipolar cells in the mouse retina. This amount depends on the set point of the thresholding in this synapse and thereby the model parameters used in the analysis presented in Figure 7 in this paper. The authors explore the impact of multiple factors in their simulation model but not this important issue. It would be important to analyze the impact of the setpoint of the nonlinearity in the modeling approach.

---

## [Author Response]

Reviewer #2:[…] The authors should provide further clarification to the following issues prior to publication:1. The authors indicate that anionic currents associated with presynaptic glutamate transport vary linearly with glutamate levels in the synaptic cleft. However, a close examination of the cited work (Hasegawa et al., 2006; Otis and Jahr, 1998) reveal several non-linear regions in the glutamate – IA(glu) dose response. Therefore, the authors should provide further evidence for linearity and ensure that their measurements (especially the large multivesicular events) are indeed within the linear region of detection.

We added some additional information for the reader to the introduction, citing data from Hasegawa et al. indicating that glutamate transporters are not saturated during rod release:

“Glutamate reuptake into rods continues even after post-synaptic mGluR6 receptors in rod bipolar cells become saturated suggesting that transporters are not saturated during rod release {Hasegawa, 2006 #1362}. In that range, glutamate anion currents (I_A(glu)_) vary linearly with glutamate levels in the synaptic cleft and can thus provide a presynaptic measure of glutamate release (Hasegawa et al., 2006; Otis and Jahr, 1998).”

2. In the manuscript, it is not clearly stated whether the resting Ca^2+^ independent release is independent of the multivesicular release seen at elevated Ca^2+^. In my reading, the data suggests that the baseline Poisson process persists even at higher membrane potentials while the multivesicular release process emerges independently. Or do the authors think the two forms of release (univesicular and multivesicular) have the same origin? Further clarification of this issue will be useful.

We agree with the reviewer that the two likely emerge independently and now say so explicitly.

Reviewer #3:[…] I have only one major comment for the authors.There has been debate on the precise amount of single-photon responses eliminated by the nonlinear thresholding in between rods and rod bipolar cells in the mouse retina. This amount depends on the set point of the thresholding in this synapse and thereby the model parameters used in the analysis presented in Figure 7 in this paper. The authors explore the impact of multiple factors in their simulation model but not this important issue. It would be important to analyze the impact of the setpoint of the nonlinearity in the modeling approach.

The reviewer wonders how the detection performance of the multi-vesicular release model (Figure 7) relates to the thresholding nonlinearity at the rod-to-rod bipolar cells synapse described in several previous studies (Field and Rieke 2002; Bernston, Smith and Taylor 2004; Trexler, Casti and Zhang 2011). The thresholding nonlinearity described in previous work has been related to the (analog) amplitude of the single photon response. Some work argues that the location of the threshold is slightly above the mean single photon response, so that fewer than half of the single photon responses make it past the nonlinearity (e.g. Field and Rieke 2002). Other work has argued that it is slightly below the mean single photon response (e.g. Bernston, Smith and Taylor, 2004) indicating that a bit more than half of the single photon responses traverse the synapse. It is also worth noting, that the locations of the nonlinearity relative to the single-photon response amplitude is almost certainly very species-dependent, as different species have different signal-to-noise ratios for their single-photon responses (Field and Rieke 2002; Trexler Casti and Zhang 2011; Field, Uzzell, Chichilnisky, and Rieke 2019).

Notice, however, that our study indicates that the output of the rod synapse translates the analog photocurrent (or photovoltage) into what is effectively a digital code of multi-vesicular release events. This analog to digital conversion is certainly a part of the ‘nonlinearity’ described in previous studies but determining how much it contributes to the thresholding at this synapse is challenging. This is because it requires understand how small fluctuations in the amplitude of the photocurrent are related to temporal variability in transmitter release. This would effectively require simultaneous photocurrent (or photovoltage) measurements with measurements of the glutamate transporter currents, and/or a full biophysical model of the rod that relates noise in the membrane voltage to variability in multi-vesicular release events. Notice that the model in Figure 7 doesn’t model the photocurrent (or photovoltage); instead, it begins with a model of the dynamics of transmitter release (see Figure 7B) and only models the observed variability in transmitter release dynamics. While understanding precisely how this analog-to-digital conversion contributes to the thresholding nonlinearity between rods and rod-bipolar cells is certainly interesting, we consider this to be beyond the scope of this study. However, we now explicitly acknowledge this issue and limitation in a new paragraph in the Discussion.